# Metformin Attenuates Monosodium-Iodoacetate-Induced Osteoarthritis via Regulation of Pain Mediators and the Autophagy–Lysosomal Pathway

**DOI:** 10.3390/cells10030681

**Published:** 2021-03-19

**Authors:** Hyun Sik Na, Ji Ye Kwon, Seon-Yeong Lee, Seung Hoon Lee, A Ram Lee, Jin Seok Woo, KyungAh Jung, Keun-Hyung Cho, Jeong-Won Choi, Dong Hwan Lee, Hong-Ki Min, Sung-Hwan Park, Seok Jung Kim, Mi-La Cho

**Affiliations:** 1The Rheumatism Research Center, Catholic Reasearch Institute of Medical Science, The Catholic University of Korea, Seoul 06591, Korea; nayoy@catholic.ac.kr (H.S.N.); jyyj7082@nate.com (J.Y.K.); seonyeong@catholic.ac.kr (S.-Y.L.); redcap817@catholic.ac.kr (S.H.L.); rveng93@catholic.ac.kr (A.R.L.); ulbojs@catholic.ac.kr (J.S.W.); chokh@catholic.ac.kr (K.-H.C.); garden7938@catholic.ac.kr (J.-W.C.); 2Department of Biomedicine & Health Sciences, College of Medicine, The Catholic University of Korea, 222, Banpo-daero, Seocho-gu, Seoul 06591, Korea; 3Impact Biotech, Korea 505 Banpo-Dong, Seocho-Ku, Seoul 06591, Korea; chu4222@gmail.com; 4Department of Orthopedic Surgery, Uijeongbu St. Mary’s Hospital, College of Medicine, The Catholic University of Korea, Seoul 11765, Korea; ldh850606@naver.com; 5Division of Rheumatology, Department of Internal Medicine, Konkuk University Medical Center, Seoul 05030, Korea; alsghdrl1921@naver.com; 6Division of Rheumatology, Department of Internal Medicine, Seoul St. Mary’s Hospital, College of Medicine, The Catholic University of Korea, Seoul 06591, Korea; 7Department of Medical Lifescience, College of Medicine, The Catholic University of Korea, Seoul 06591, Korea

**Keywords:** osteoarthritis, metformin, pain, cartilage, autophagy, combination therapy

## Abstract

Osteoarthritis (OA) is the most common degenerative arthritis associated with pain and cartilage destruction in the elderly; it is known to be involved in inflammation as well. A drug called celecoxib is commonly used in patients with osteoarthritis to control pain. Metformin is used to treat type 2 diabetes but also exhibits regulation of the autophagy pathway. The purpose of this study is to investigate whether metformin can treat monosodium iodoacetate (MIA)-induced OA in rats. Metformin was administered orally every day to rats with OA. Paw-withdrawal latency and threshold were used to assess pain severity. Cartilage damage and pain mediators in dorsal root ganglia were evaluated by histological analysis and a scoring system. Relative mRNA expression was measured by real-time PCR. Metformin reduced the progression of experimental OA and showed both antinociceptive properties and cartilage protection. The combined administration of metformin and celecoxib controlled cartilage damage more effectively than metformin alone. In chondrocytes from OA patients, metformin reduced catabolic factor gene expression and inflammatory cell death factor expression, increased LC3Ⅱb, p62, and LAMP1 expression, and induced an autophagy–lysosome fusion phenotype. We investigated if metformin treatment reduces cartilage damage and inflammatory cell death of chondrocytes. The results suggest the potential for the therapeutic use of metformin in OA patients based on its ability to suppress pain and protect cartilage.

## 1. Introduction

Osteoarthritis (OA) is a type of chronic and degenerative arthritis and the most common continuous joint disorder in the elderly. The severe pain and irreversible cartilage damage associated with OA reduces the quality of life of affected patients [1]. Thus, the treatment of OA is aimed at relieving arthralgia and preventing disease progression, including cartilage degradation, which is irreversible and may aggravate arthritic pain. However, effective disease-modifying osteoarthritis drugs that prevent cartilage damage in OA are not yet available [2]. While OA has traditionally been considered noninflammatory arthritis, recent studies have shown that its pathogenesis includes low-grade inflammation [3] and the involvement of numerous proinflammatory cytokines and matrix metalloproteinases (MMPs) in its aggravation [4]. Targeting these proinflammatory cytokines has thus emerged as a therapeutic option in OA [4]. The destruction of bone and cartilage gives rise to a state of chronic inflammation [5] and is consistent with the finding of a smaller population of differentiated regulatory T (Treg) cells in the synovial membranes of OA vs. RA patients [6]. Interleukin (IL)-10 expression by Treg cells is also reduced in OA patients [7], whereas levels of helper T (Th)17 cells in their joints are increased [8]. This imbalance of Th17/Treg, and therefore of pro- and anti-inflammatory cytokines, may contribute to the pathogenesis of OA.

The biguanide metformin has been widely used for the treatment of type 2 diabetes [9]. However, studies of experimental animal models of autoimmune arthritis have shown that metformin also has antiarthritic activity and inhibits bone degradation [10,11]. In other reports, metformin restored the balance between Th17 and Treg cells in an animal model of inflammation [12] and decreased both osteoclastogenesis and expression of MMP-9 in vitro [13].

Recent studies suggest the potential use of metformin for OA therapy. Metformin attenuated cartilage damage and controlled pain in a destabilization of the medial meniscus (DMM) model of OA in mice [14]. Moreover, metformin protected chondrocytes against IL-1β-induced inflammatory response and injury [15]. We also showed the therapeutic effects of metformin in a previous study [16]. OA rats treated with metformin-stimulated adipose-tissue-derived human MSCs (Ad-hMSCs) experienced increased antinociceptive activity and chondropotective effects.

Celecoxib is known as a COX-2 inhibitor and nonsteroidal anti-inflammatory drug (NSAID). It has regulatory effects of pain and inflammation in osteoarthritis [17]. In a cohort study, the rate of joint replacement surgery was lower in OA patients with type 2 diabetes mellitus when they were treated with a combination of metformin and celecoxib. The results suggest the possibility of the therapeutic use of metformin and celecoxib combined on OA patients [18].

We, therefore, hypothesized that metformin could ameliorate OA via antinociceptive and anti-inflammatory effects. Thus, in this study, we investigated the therapeutic effect and the related mechanism of action of metformin in both an animal model of OA and in chondrocytes from OA patients. In addition, the effects of metformin in OA patients were evaluated by monitoring radiographically evaluated disease progression in the knees of diabetic OA patients treated with metformin.

## 2. Materials and Methods

### 2.1. Animals

Seven-week-old male Wistar rats weighing 180–250 g at the start of the experiment were purchased from Central Lab Animal Inc. (Seoul, South Korea). A maximum of three animals per cage was housed in a room with controlled temperature (20–26 °C) and light (12-h light–dark cycle) conditions. The rats had free access to a gamma-ray-sterilized diet (TD 2018S, Envigo (Indianapolis, IN, USA) and autoclaved R/O water. All animal research procedures were conducted in accordance with the Laboratory Animals Welfare Act, the Guide for the Care and Use of Laboratory Animals, and the Guidelines and Policies for Rodent Experiments provided by the Institutional Animal Care and Use Committee (IACUC) of the School of Medicine, The Catholic University of Korea. The IACUC and the Department of Laboratory Animals of the Catholic University of Korea, Songeui Campus, accredited the Korean Excellence Animal Laboratory Facility in accordance with the Korean Food and Drug Administration in 2017, and full accreditation by AAALAC International was acquired in 2018.

### 2.2. Induction of Osteoarthritis and Treatment with Metformin

Animals were randomly assigned to the treatment or control groups before the study began. After anesthetization with isoflurane, the Wistar rats of seven-week-old males (n = 6) were injected with 3 mg of monosodium iodoacetate (MIA; Sigma, St. Louis, MO, USA), dissolved in a 50-μL volume, using a 26.5-G needle inserted through the patellar ligament into the intra-articular space of the right knee. Metformin 100 mg/kg was orally administered to the MIA-induced rats using an oral gavage needle for 14 days. In addition, the metformin and celecoxib complex was administered at a concentration of metformin 50 mg/kg and celecoxib 80 mg/kg.

### 2.3. Assessment of Pain Behavior

Nociception in MIA-treated rats, randomized to the different experimental groups, was tested using a dynamic plantar aesthesiometer (Ugo Basile, Gemonio, Italy). The device is an automated version of the von Frey hair assessment procedure and is used to assess mechanical sensitivity. The assessment was conducted by placing the rats on a metal mesh surface in an acrylic chamber in a temperature-controlled room (20–26 °C), where they rested for 10 min before the touch stimulator unit was positioned under each animal. An adjustable angled mirror was used to place the stimulating microfilament (0.5 mm diameter) below the plantar surface of the hind paw. When the instrument was activated, a fine plastic monofilament advanced at a constant speed and touched the paw in the proximal metatarsal region. The filament exerted a gradually increasing force on the plantar surface, starting below the threshold of detection and increasing until the stimulus became painful, as indicated by the rat’s withdrawal of its paw. The force required to elicit a paw-withdrawal reflex was recorded automatically and measured in g. A maximum force of 50 g and a ramp speed of 25 s were used for all aesthesiometer tests.

### 2.4. Assessment of Weight-Bearing

Weight balance in MIA-treated rats was analyzed using an incapacitance meter (IITC Life Science, CA, USA). The rats were allowed to acclimate for 5 min in an acrylic holder. After 5 min, both feet of the rat were fixed to the pad and the weight balance was measured for 5 s. Three measurements were repeated in the same manner. The weight of the unguided and guided legs was determined and substituted into the formula to find the% value. The% value was obtained by comparing the leg with and the leg without osteoarthritis.

### 2.5. Histopathological Analysis

Knee joints and dorsal root ganglions were collected from each group at 2 weeks post-MIA induction. The tissues were fixed in 10% formalin solution, decalcified using Decalcifying Solution-Lite (Sigma, St. Louis, MO, USA), and embedded in paraffin. Sections of 4- to 5-μm thickness were cut, dewaxed using xylene, dehydrated through an alcohol gradient, and then stained with hematoxylin and eosin (H&E) and safranin O. Stained samples were analyzed by Osteoarthritis Research Society International (OARSI) and Mankin scoring systems [19].

### 2.6. Immunohistochemistry

Paraffin-embedded sections were incubated at 4 °C with the following primary monoclonal antibodies: anti-transient receptor potential cation channel subfamily V member 1 (TRPV-1; 1:400 dilution, Cat no. AF3066, R&D Systems, Minneapolis, MN, USA), anti-calcitonin gene-related peptide (CGRP; 1:400 dilution, Cat no. ab81887, Abcam, Cambridge, UK), anti-IL-1β (1:500 dilution, Cat no. ab9722, Abcam), anti-IL-17 (1:500 dilution, Cat no. ab79056, Abcam), anti-MMP-3 (1:50 dilution, Cat no. ab52915, Abcam), anti-inducible nitric oxide synthase (iNOS; 1:100 dilution), anti-caspase-1 (1:400 dilution, Cat no. ab108362, Abcam), anti-phospho-mixed lineage kinase domain-like protein (MLKL; 1:150 dilution, Cat no. ab196436, Abcam), and anti-phospho-AMP-activated protein kinase (AMPK; 1:500 dilution, Cat no. 2535S, Cell Signaling Technologies, Danvers, MA, USA). The samples were then incubated with the respective secondary biotinylated antibodies, followed by 30-min incubation with a streptavidin–peroxidase complex. The reaction product was developed using 3,3-diaminobenzidine chromogen (Dako, Carpinteria, CA, USA).

### 2.7. Human Articular Chondrocyte Differentiation and Culture

OA patients were recruited from the Orthopedic Surgery, Uijeongbu St. Mary’s Hospital, Seoul, Korea (IRB No. UC18RESI0038). Articular cartilage from humans was acquired from patients undergoing replacement arthroplasty or joint replacement surgery. Cartilage obtained from patients was digested and reacted with 0.5 mg/mL hyaluronidase, 5 mg/mL protease type XIV, and 2 mg/mL collagenase type V. Finally, chondrocytes were incubated in Dulbecco’s modified Eagle medium (DMEM), including 10% fetal bovine serum. The isolated human OA chondrocytes of passage 3 were cultured with metformin (1 mM) and IL-1β (20 ng/mL) for 24 h.

### 2.8. Real-Time Polymerase Chain Reaction (PCR)

RNA was extracted using TRIzol reagent (Molecular Research Center, Inc., Cincinnati, OH, USA). cDNA was synthesized using the Superscript Reverse Transcription System (TaKaRa, Shiga, Japan), and quantitative real-time polymerase chain reaction (PCR) was performed using LightCycler FastStart DNA Master SYBR Green I (TaKaRa) according to the manufacturer’s instructions.

The primer sequences for PCR were designed using Primer Express (Applied Biosystems, Foster City, CA, USA) and were as follows; *MMP-1*, 5′-CTG AAG GTG ATG AAG CAG CC-3′ (sense) and 5′-AGT CCA AGA GAA TGG CCG AG-3′ (antisense); for *MMP-3*, 5′-CTC ACA GAC CTG ACT CGG TT-3′ (sense) and 5′-CAC GCC TGA AGG AAG AGA TG-3′ (antisense); for *MMP-13*, 5′-CTA TGG TCC AGG AGA TGA AG-3′ (sense) and 5′-AGA GTC TTG CCT GTA TCC TC-3′ (antisense); for *TIMP1*, 5′-AAT TCC GAC CTC GTC ATC AG-3′ (sense) and 5′-TGC AGT TTT CCA GCA ATG AG-3′ (antisense); for *TIMP3*, 5′-CTG ACA GGT CGC GTC TAT GA-3′ (sense) and 5′-GGC GTA GTG TTT GGA CTG GT-3′ (antisense); for *AMPKα1*, 5′-AAC TGC AGA GAG CCA TTC ACT TT-3′ (sense) and 5′-GGT GAA ACT GAA GAC AAT GTG CTT-3′ (antisense).

### 2.9. Western Blot Analysis

Chondrocytes (1 × 10^6^ cell/well) were incubated with IL-1β 20 ng/mL (R&D Systems, MN, USA) and metformin 0.2, 1, and 5 mM (Sigma Aldrich) for 1 h. Proteins were separated by SDS-PAGE and transferred onto a nitrocellulose membrane (Amersham Pharmacia Biotech, Piscataway, NJ, USA). Western blotting was performed using a SNAP i.d. protein detection system (Millipore). The hybridized bands were detected using an enhanced chemiluminescence (ECL) detection kit (Thermo Fisher Scientific, MA. USA) and the following antibodies: anti-phospho-AMPK (1:1000 dilution), anti-AMPK (1:1000 dilution, Cat. No. #2535), anti-LC3B (1:2000 dilution, Cat. No. #2775), anti-p62 (1:400 dilution, Cat. No. #5114), anti-caspase-1 (1:1000 dilution, Cat. No. #2225), anti-caspase-3 (1:1000 dilution, Cat. No. #9662) (all from Cell Signaling Technologies), anti-GAPDH (1:2000 dilution, Cat. No. ab9485, Abcam), and anti-rabbit IgG-HRP (1:2000 dilution, Cat. No. sc2357, Santa Cruz Biotechnology, Dallas, Texas, USA). The Western blot bands were quantified using the Fiji/ImageJ program.

### 2.10. In Vivo Microcomputed Tomography (CT) Iimaging and Analysis

Micro-CT imaging and analysis were performed using a bench-top cone-beam type in vivo animal scanner (mCT 35; SCANCO Medical, Bruttisellen, Switzerland). The animals were imaged at settings of 70 kVp and 141 μA using an aluminum 0.5-mm thick filter. The pixel size was 8.0 μm, and the rotation step was 0.4 °C. Cross-sectional images were reconstructed using a filtered back-projection algorithm (NRecon software, Bruker micro CT, Kontich, Belgium). For each scan, a stack of 286 cross-sections was reconstructed at 2000 × 1335 pixels. Bone volume and surface were analyzed at the femur.

### 2.11. Immunofluorescence

Human chondrocytes of passage 3 were cultured with metformin (1 mM) and IL-1β (20 ng/mL) for 24 h. The cells were stained with specific Abs for autophagosome and autophagolysosome. The cells were stained with BODIPY (Cat. No. #D3922, Thermo Fisher, USA) for lipid droplets and PE-conjugated anti-LC3b (Cat. No. #sc-20011, Santa Cruz Biotechnology) and APC-conjugated anti-LAMP1 (Santa Cruz Biotechnology) for autophagy, and DAPI for nucleus. Fluorescence analysis was performed using the ZEN2012 (blue edition; ZEISS) program. The positive color was analyzed against the background value by dividing the sites of the photograph.

### 2.12. Grading System of Kellgren and Lawrence (KL) in OA Patients

The radiographs of both knees of OA patients treated with NSAIDs for >3 years were evaluated retrospectively. The data from 41 patients, divided into two groups—diabetic patients who had taken metformin to control their blood glucose level and nondiabetic patients, were collected and analyzed (IRB. No. UC18RESI0038).

### 2.13. Statistical Analysis

Statistical analysis was performed using GraphPad Software (version 5.01, San Diego, CA, USA). Statistical significance of data with multiple groups was calculated using one-way ANOVA; if a significant difference was observed among groups, the Bonferroni posthoc test was used to assess the statistical difference between specific groups. Comparisons of numerical data between two groups were performed with a nonparametric Mann–Whitney U-test (two-tailed). All of the data are presented as the mean ± standard error of the mean. *p*-values of 0.05 were considered significant.

## 3. Results

### 3.1. Metformin Reduces Pain in MIA-Induced OA Rats

The ability of metformin to attenuate pain in MIA-induced OA rats was assessed based on secondary tactile allodynia. Compared to control MIA rats, metformin administration increased paw-withdrawal latency (PWL), paw-withdrawal threshold (PWT), and weight-bearing, as determined in an automated von Frey hair assessment (Figure 1A,B). In addition, expression of CGRP in dorsal root ganglia (DRG) was decreased in the metformin-treated group compared to the control group, but not TRPV1 (Figure 1C). These results imply that metformin can attenuate OA-related pain by inhibiting nociception and pain severity.

### 3.2. Protective Effects of Metformin in the Knee Joints of MIA-Induced OA Rats

Quantitative micro-CT showed that metformin protected the femurs of MIA-induced OA rats (Figure 2A), as shown by the increases in bone volume and bone surface in the treated group vs. the control group (Figure 2B). Safranin O staining of the femur revealed that metformin decreased cartilage destruction of the knee joint (Figure 2C). Additionally, metformin administration reduced both OARSI and total Mankin scores compared to the control group (Figure 2D). Together, these data confirm that metformin slows OA progression by reducing bone and cartilage damage during OA development.

### 3.3. Metformin Reduces the Levels of Inflammatory Mediators and Catabolic Factors in the Synovium of OA Rats

Immunohistochemistry was used to detect inflammatory mediators and the production of catabolic factors in the joints of OA rats. In the metformin-treated group, expression of MMP-3, IL-1β, iNOS, and IL-17 was reduced in OA synovium tissues (Figure 3A). These results imply that metformin limits catabolic and inflammatory responses in MIA-induced OA rats.

### 3.4. Metformin Decreases the Catabolic Response of Human OA Chondrocytes

Real-time PCR was conducted to evaluate the therapeutic function of metformin in IL-1β-stimulated human OA chondrocytes. Expression of the genes encoding MMP-1, -3, and -13 was downregulated in metformin-treated vs. vehicle-treated cultures (Figure 3B), whereas TIMP-1 and-3 mRNA levels were increased (Figure 3C). An increase in AMPK, which is activated by metformin and leads to decreased expression of procatabolic factors in chondrocytes [20], was also observed (Figure 3D). These results imply that the mechanisms underlying OA prevention by metformin include control of the reciprocal balance between anabolic and catabolic responses and enhanced AMPK activation.

### 3.5. Metformin Regulates Inflammation-Induced Cell Death in the Joints of MIA-Induced OA Rats

We investigated whether metformin can reduce inflammatory cell death through autophagy in OA. Immunohistochemistry was used to detect autophagy and the levels of cell death mediators in MIA-treated rat joints. Expression of AMPK, which is also an activator of autophagy, was significantly increased by metformin administration, whereas both caspase-1 and p-MLKL levels were significantly reduced (Figure 4A). Western blotting revealed increases in the expression of caspase-3 and p-AMPK, LC32b, and p62, while caspase-1 was decreased in metformin-treated chondrocytes (Figure 4B,C). In addition, the inflammatory cell death factor, caspase-1, was increased by the treatment of hydroxychloroquine (HCQ), which is an autophagy inhibitor (Figure 4D). These results suggest that metformin regulates inflammatory cell death through inhibition of autophagy in OA chondrocytes.

### 3.6. Metformin Promotes Autophagy in Human Chondrocytes

Human OA chondrocytes stimulated by IL-1β were examined for autophagolysosome markers by immunofluorescence staining. The number of LC3B- and LAMP1-positive cells increased in the metformin-treated group compared with the IL-1β-stimulated group (Figure 5A,B). These findings indicate that metformin regulates the expression of autophagolysosome markers in chondrocytes.

### 3.7. The Effect of the Combination of Metformin and Celecoxib in OA Progression

It is well known that celeccoxib has a therapeutic effect in OA [21]. In this study, we investigated the effect of metformin and celecoxib combined on OA progression. To address that, we administrated a combination of metformin and celecoxib to MIA-induced OA rats. Cartilage damage was dramatically improved in the metformin and celecoxib combination-treated group (Figure 6A,B). This result shows the therapeutic potential of cotreatement of metformin and celecoxib in OA.

### 3.8. Metformin Inhibits Clinical-Grade OA and Exhibits an Anti-Inflammatory Effect in OA Patients

The radiographs of both knees of OA patients treated with celecoxib for >3 years were evaluated retrospectively. Sixty patients were divided into two groups: diabetic patients who had taken metformin to control their blood glucose level and nondiabetic patients. The ability of metformin to inhibit the clinical symptoms of OA was evaluated based on the Kellgren and Lawrence (KL) grade [22]. The results showed a lower mean KL grade in the metformin group than in the non-metformin group (Figure 7).

## 4. Discussion

The antiarthritic and anti-inflammatory effects of metformin have been reported [12,13]. It has been shown that metformin has a therapeutic effect on OA animal models and cohort studies [14,18,23]. The present study confirmed the therapeutic effect of metformin in rats with MIA-induced OA and, thus, its potential clinical use on OA patients. Specifically, we showed that metformin slows OA progression by reducing pain mediators and cartilage damage. A previous study recommended the use of anti-inflammatory agents in the treatment of OA based on the low-grade chronic inflammatory response associated with the pathogenesis of the disease and the resulting bone deformation [24]. Indeed, blocking IL-6 production prevents OA progression in animal models, while NSAIDs suppress inflammatory cytokine levels in OA synovial fluid [25,26]. Metformin seems to have not only an antiglycemic effect but also an anti-inflammatory effect [27]. In the present study, administration of metformin prevented the progression of MIA-induced OA in rats and was associated with a lower KL grade in the knees of diabetic OA patients. The underlying mechanisms included inhibition of cartilage destruction, decrease of proinflammatory cytokines and catabolic enzymes, and amelioration of pain severity.

In the pathogenesis of OA, catabolic activity results in the irreversible degradation of bone and cartilage, which further contributes to the pain level. Catabolic mediators, such as MMPs, have been shown to exacerbate OA severity by promoting cartilage damage [28], whereas TIMPs suppress MMP activity and maintain cartilage [29,30]. The balance between anabolic and catabolic activities is thought to play an important role in regulating cartilage and tissue repair in OA patients [31]. In addition, metformin, as an inducer of AMPK signaling, induces cartilage maintenance by suppressing inflammation-induced cartilage degradation [23]. A recent study showed smaller Th17 and Treg populations in the peripheral blood of OA vs. RA patients [32]. We found that metformin induced the activation of AMPK as well as reciprocal regulation of both MMPs and TIMPs. Moreover, metformin protects chondrocytes from inflammatory cell death by reducing the IL-1β-induced inflammatory response [15]. Our previous study also showed that metformin has a therapeutic effect in experimental OA [16]. These results imply the possibility of using metformin as a disease-modifying OA drug, based on its ability to regulate the immunologic and inflammatory/catabolic pathways. Clinically, OA patients suffer considerable pain, which, in addition to severe disabilities, results in significant socioeconomic costs [33]. Proinflammatory cytokines, such as IL-6 and tumor necrosis factor (TNF)-α, have been shown to sensitize nociceptors in rats [34,35], whereas TNF-α blockade attenuated pain behaviors and joint destruction in a mouse model of arthritis [36]. Both TRPV1 and CGRP have been implicated in the pain response [37,38], while the inhibition of both was shown to improve metabolic health during aging [38]. In the present study, metformin decreased pain severity by increasing PWT and PWL and improved weight-bearing in rats with MIA-induced OA. Metformin also decreased the expression of TRPV1 and CGRP in the DRG of OA rats. These results imply that metformin relieves pain through the nociceptive pathway in OA.

Autophagy is an essential cellular and molecular mechanism that is important for maintaining cellular homeostasis [39], and it has been reported that autophagy is reduced in the cartilage of OA patients [40]. Recently Wang and colleagues reported the protective effects of metformin against osteoarthritis through the activation of autophagy [41,42]. In this study, our result confirmed the therapeutic effects of metformin through the activation of autophagy.

Inflammatory cell death, including pyroptosis and necroptosis, is induced by impairment of the autophagy process, which is triggered by the inflammation response. In osteoarthritis, it is well known that the chondrocytes are susceptible to apoptosis [43]. A recent report has shown that necrosis occurs first, followed by apoptosis in chondrocytes [44]. We investigated whether metformin can reduce inflammatory cell death through autophagy formation in OA chondrocytes. Metformin induced autophagy formation through p-AMPK, LC3IIB, and p62 activation. In addition, inflammation cell death factors, p-MLKL and caspase 1, are decreased by metformin treatment in OA chondrocytes.

A recent study showed the chondroprotective effect of metformin in injury-induced OA animal models [23], which is consistent with our result. OA can have multiple causes, and metformin has a tentative effect when OA has mechanical causes such as meniscus surgery. Although the study proved the role of metformin in injury-induced OA, there are some differences with our study.

First, we used the MIA-induced OA model, which is more similar to actual OA, with a DM (diabetes mellitus) patient group, in which OA occurs without meniscus tear or ligament injury. It is considered a more suitable model for the clinical application of metformin in the future.

Second, the effect of metformin can be confirmed by following the radiographs of diabetic patients taking NSAIDs (celecoxib) and metformin together and the radiographs of nondiabetes OA patients taking only NSAIDs (celecoxib) for 3 years. In other words, the combination of metformin and celecoxib was effective in inhibiting the development of osteoarthritis [18].

In conclusion, this study demonstrates the therapeutic effect of metformin in OA by mechanisms that include cartilage protection and pain relief. Therefore, metformin should be considered a potential therapeutic candidate for OA.

## 5. Conclusions

We evaluated the ability of metformin to ameliorate the progression of osteoarthritis. Metformin ameliorated the progression of experimental OA and exhibited both antinociceptive properties and cartilage protection. Metformin-induced downregulation of pain mediators, including CGRP, in dorsal root ganglia and protection against bone damage were also determined. In chondrocytes from OA patients, metformin decreased the expression of genes encoding catabolic factors stimulated by interleukin-1β. In addition, metformin regulated inflammatory cell death, such as necroptosis, and induced the autophagosome–lysosome fusion phenotype. The combination therapy of metformin and celecoxib had significantly less cartilage damage than either metformin alone or the control. Our results suggest that the possibility of the therapeutic use of metformin in OA patients, based on its ability to suppress pain and protect cartilage [18].

## Figures and Tables

**Figure 1 cells-10-00681-f001:**
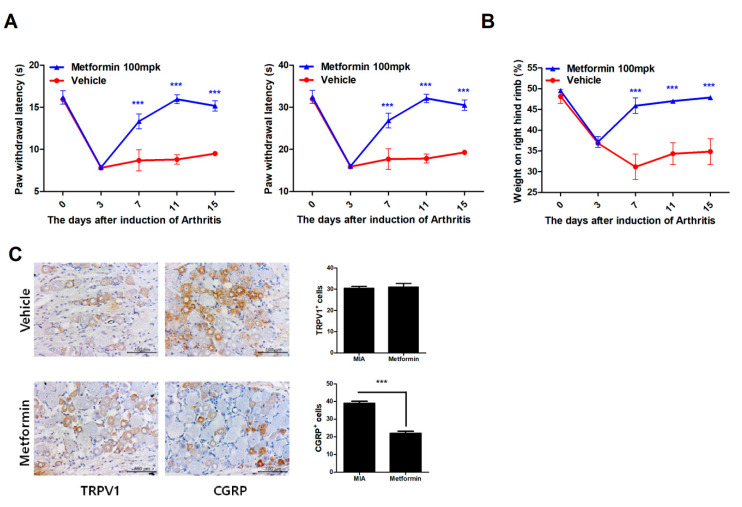
Metformin reduces pain in monosodium iodoacetate (MIA)-induced OA rats. The rats were injected with 3 mg of MIA. After osteoarthritis (OA) induction, the rats were administrated with metformin 100 mg/kg (N = 6 animals per group). (**A**) Nociceptive testing was performed using a dynamic plantar aesthesiometer (Ugo Basile, Gemonio, Italy), an automated version of the von Frey hair assessment procedure. (**B**) Weight-bearing was assessed using an incapacitance meter (IITC Life Science, Woodland Hills, CA, USA). (**C**) Dorsal root ganglion tissue from rats in all groups was stained immunohistochemically with specific antibodies for CGRP and TRPV1. Immunohistochemically identified TRPV1- and CGRP-positive cells were counted. Data are represented as the mean ± standard error of the mean (SEM) of three independent experiments (*** *p* < 0.001).

**Figure 2 cells-10-00681-f002:**
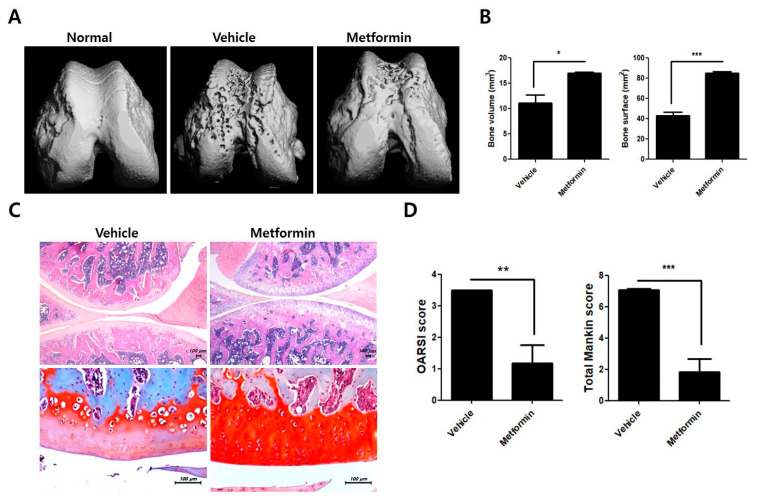
Metformin ameliorates bone and cartilage erosion in MIA-induced OA rats. The rats were injected with 3 mg of MIA. After OA induction, the rats were administrated with metformin 100 mg/kg (N = 6 animals per group). The rats were sacrificed at Day 15 after OA induction to collect joint tissues. (**A**) Twenty cylindrical bone samples were obtained from bone biopsies of 20 dry hemimandibles. The samples of six animals per group were scanned using microcomputed tomography (mCT 35; SCANCO Medical, Zurich, Switzerland). (**B**) Bone volume and bone surface were analyzed using NRecon software. (**C**) Knee-joint tissue samples were acquired from all OA groups at 2 weeks and stained with hematoxylin and eosin (H&E) and safranin O (**D**) to determine Osteoarthritis Research Society International (OARSI) and Mankin scores. Data are represented as the mean ± SEM of three independent experiments (* *p* < 0.05, ** *p* < 0.01, *** *p* < 0.001).

**Figure 3 cells-10-00681-f003:**
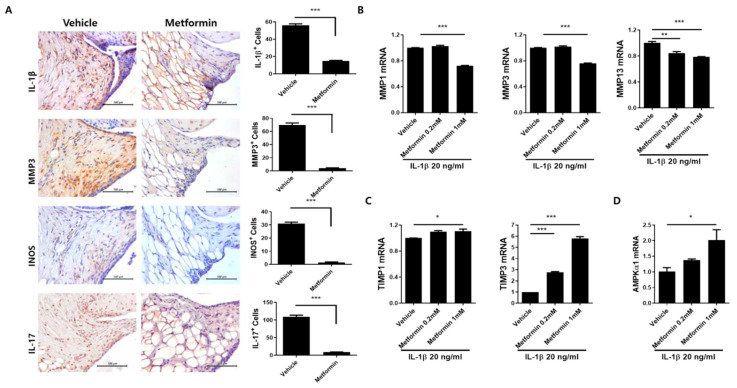
Expression of Inflammatory mediators following metformin administration. The rats were injected with 3 mg of MIA. After OA induction, the rats were administrated with metformin 100 mg/kg (N = 6 animals per group). The OA rats were sacrificed at Day 15 after OA induction to collect joint tissues. (**A**) The expression of IL-1β, MMP3, iNOS, and IL-17 after metformin administration was determined immunohistochemically in the synovium of rats with MIA-induced OA. The count of positive cells for each antibody is shown on the right. The data are reported as the mean ± SD of three independent experiments, with six animals per group. *** *p* < 0.001. (**B**) The human OA chondrocytes were cultured with metformin (0.2 and 1 mM) and IL-1β (20 ng/mL) for 24 h. mRNA levels for MMP-1, -3, and -13, (**C**) TIMP-1, -3, and (**D**) AMPKα1 were measured by real-time-PCR. Data are represented as the mean ± SEM of three independent experiments (* *p* < 0.05, ** *p* < 0.01, *** *p* < 0.001).

**Figure 4 cells-10-00681-f004:**
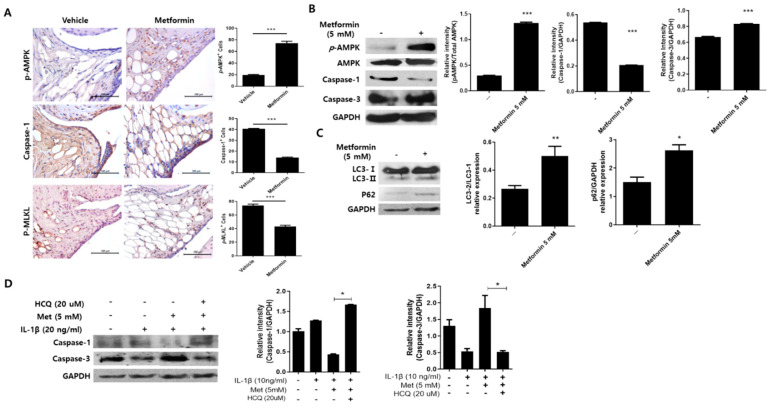
Expression of cell death mediators following metformin administration. (**A**) The expression of p-AMPK, caspase-1, and p-MLKL in the knee joint tissues of MIA-induced OA rats was determined immunohistochemically (N = 6 animals per group). The count of positive cells for each antibody is shown on the right. Human OA chondrocytes were cultured with metformin (5 mM) for 24 h. (**B**,**C**) p-AMPK, caspase-1, caspase-3, LC3, p62, and GAPDH were analyzed using Western blot in the protein of metformin-treated human chondrocytes. (**D**) The human OA chondrocytes were cultured with metformin (5 mM), IL-1β (20 ng/mL), and HCQ (20 μM) for 24 h. Caspase-1, caspase-3, and GAPDH were analyzed using Western blot. Data are represented as the mean ± SEM of three independent experiments (* *p* < 0.05, ** *p* < 0.01, *** *p* < 0.001).

**Figure 5 cells-10-00681-f005:**
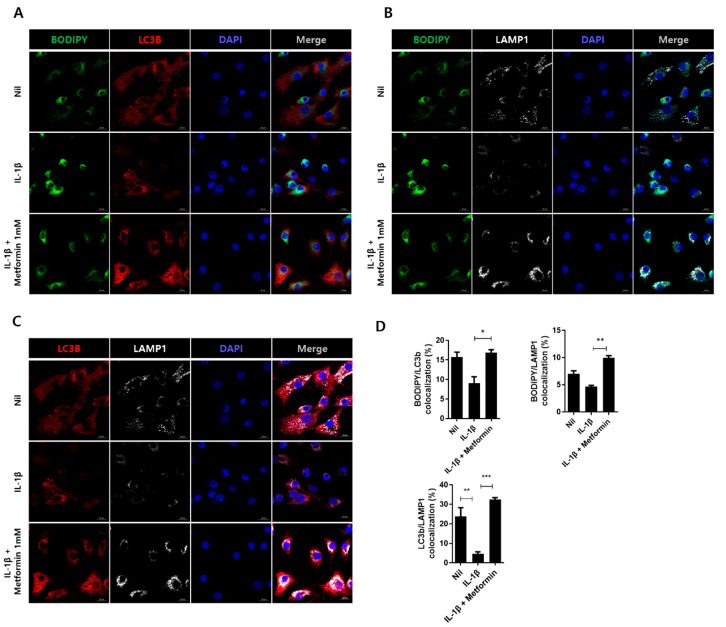
Metformin regulates autophagolysosome via LAMP1 activation. Human chondrocytes were cultured with metformin (1 mM) and IL-1β (20 ng/mL) for 24 h. The cells were analyzed using confocal microscopy for autophagosome and autophagolysosome. For autophagosome structure analysis, (**A**) colocalization of BODIPY (green) and LC3B (red) were analyzed in unstimulated (Nil) or each stimulated human chondrocyte (**B**,**C**) for autophagolysosome, colocalization of BODIPY (green), and LAMP1 (white) or colocalization of LC3B (red) and LAMP1 (white). (**D**) Colocalization fluorescence was analyzed using the Fiji/ImageJ program. The data are a repeat of three independent experiments, and the fluorescent data are presented as the mean ± SEM of three independent experiments (* *p* < 0.05, ** *p* < 0.01, *** *p* < 0.001).

**Figure 6 cells-10-00681-f006:**
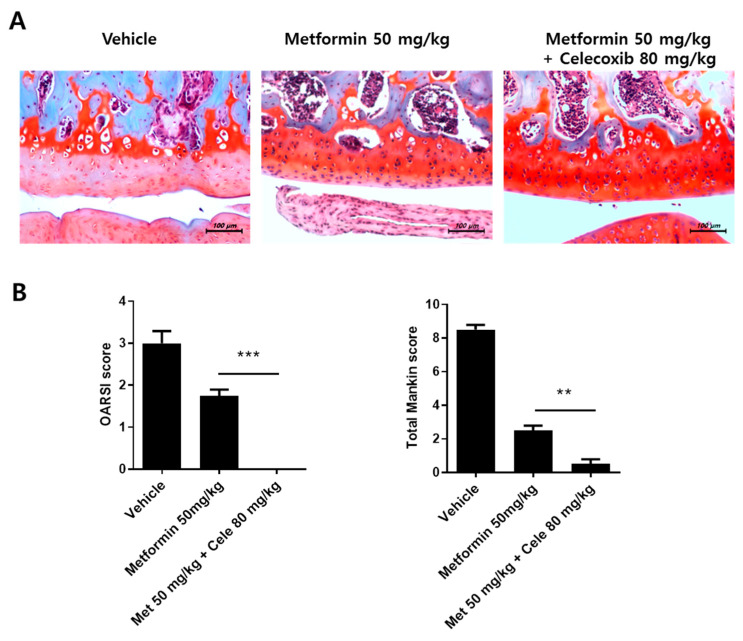
Combined effects of metformin and celecoxib. (**A**) OA was induced in Wistar rats by intra-articular injection of MIA. Metformin (50 mg/kg) and celecoxib (80 mg/kg) were orally administrated into OA rats for 14 days (N = 6 animals per group). Knee-joint tissue samples acquired from all OA groups at 14 days were stained with H&E and safranin O (**B**) to evaluate disease severity based on OARSI and Mankin scores. Data are represented as the mean ± SEM of three independent experiments (** *p* < 0.01, *** *p* < 0.001).

**Figure 7 cells-10-00681-f007:**
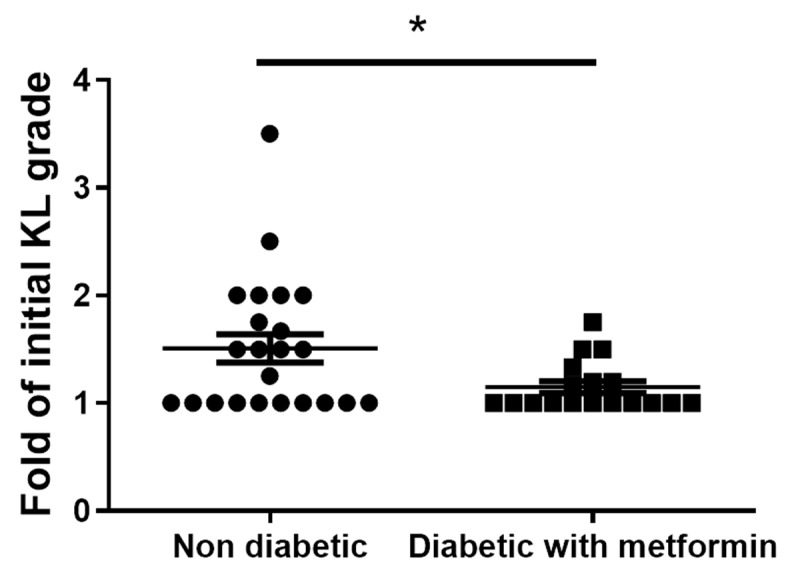
Clinical study of the effect of metformin in OA patients. A clinical assessment of OA patients with type 2 diabetes, who had been followed-up for >3 years, showed slower progression of OA in those taking metformin (nondiabetic = 23 patients, diabetic with metformin = 18 patients; all patients female). The OA grade was analyzed based on 3 years of data (* *p* < 0.05).

## Data Availability

Not applicable.

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
