# Peer review of "Metformin Attenuates Monosodium-Iodoacetate-Induced Osteoarthritis via Regulation of Pain Mediators and the Autophagy–Lysosomal Pathway"

_cells, 2021, doi:10.3390/cells10030681_

Round 1

Reviewer 1 Report

Title : Replace « improves » by « attenuates »

Abstract :

L24 (1) Background at the start of the introduction

L26: the term « ameliorate »  is ambiguous in this context: replace by « diminish »

L32 : replace also « ameliorated » by « decreased »

L34-35 : « the metformin-induced doqwregulation of pain mediators….were also determined. » Reformulate the sentence to give only the results in this section (results).

Introduction :

L67 : Give a reference : For exemple :

Sanchez-Rangel, Elizabeth, et Silvio E. Inzucchi. « Metformin: Clinical Use in Type 2 Diabetes ». Diabetologia 60, no 9 (septembre 2017): 1586‑93. https://doi.org/10.1007/s00125-017-4336-x.

L73 : The bibliography on the effects of metformin and OA is not comprehensive enough.

You should include some recent findings from studies of metformin and osteoarthritis and metformin/OA/diabetes, in particular:

Lu, Chieh-Hua, et al. « Combination COX-2 Inhibitor and Metformin Attenuate Rate of Joint Replacement in Osteoarthritis with Diabetes: A Nationwide, Retrospective, Matched-Cohort Study in Taiwan ». PloS One 13, no 1 (2018): e0191242. https://doi.org/10.1371/journal.pone.0191242.

Li, Hui, Xiang Ding, , et al. « Exploration of Metformin as Novel Therapy for Osteoarthritis: Preventing Cartilage Degeneration and Reducing Pain Behavior ». Arthritis Research & Therapy 22, no 1 (22 février 2020): 34. https://doi.org/10.1186/s13075-020-2129-y.

Sanchez-Rangel, Elizabeth, et Silvio E. Inzucchi. « Metformin: Clinical Use in Type 2 Diabetes ». Diabetologia 60, no 9 (septembre 2017): 1586‑93. https://doi.org/10.1007/s00125-017-4336-x.

You also should explain why you studied anti-inflammatory effects of metformin related to yours recent results and results of others (anti-IL-1 effects)

Park, Min-Jung, Su-Jin Moon, Jin-Ah Baek, Eun-Jung Lee, Kyung-Ah Jung, Eun-Kyung Kim, Da-Som Kim, et al. « Metformin Augments Anti-Inflammatory and Chondroprotective Properties of Mesenchymal Stem Cells in Experimental Osteoarthritis ». Journal of Immunology (Baltimore, Md.: 1950) 203, no 1 (1 juillet 2019): 127‑36. https://doi.org/10.4049/jimmunol.1800006.

Zhang, Mengqi, Yaping Liu, Zhikun Huan, Yan Wang, et Jin Xu. « Metformin Protects Chondrocytes against IL-1β Induced Injury by Regulation of the AMPK/NF-κ B Signaling Pathway ». Die Pharmazie 75, no 12 (1 décembre 2020): 632‑36. https://doi.org/10.1691/ph.2020.0762.

You should introduce celecoxib.

METHODS :

L110 : Why did you change the dose of metformin (50 mg/kg)  in the metormin/celecoxib combination?

L144 : Explain the determination of OARSI and Mankin scores, or give references.

L147 : Give the dilutions of antibodies, definition of MLK1 and AMPK

L158 : Cartilage with a capital letter. Give the approval number for human cartilage samples

L161 :  Chondrocytes without a capital letter ; Add « in » : chondrocytes were incubated in Dulbecco’s…

L187 : Give the dilutions of antibodies

L204 : Define KL : grading system of Kellgren and Lawrence, I suppose.

L201 : Immunofluorescence : The methodology is not comprehensive enough : type of culture, method, incubation, dilution of antibodies, microscope…

I don’t understand FITC-conjugated anti BODIPY because BODIPY is a dye or a tracer of lipids but not a protein. Reformulate. You realized a nuclear staining in the fig 5 (in merge), but you did’nt explain it !  (DAPI staining or another ?)

Results :

L218 : Replace « metformin suppresses pain «  by «  Metformin reduces pain »

L219 : Replace « to improve » by « to attenuate »

Figure 1 : L230-232 : Review the sentence ; L234 : Add antibodies to CGRP and TRPV1.

Specify the duration of OA induction. B is missing in the pannel at the top right.

L240, L251, Fig2B : What do you mean by bone surface ? if cartilage, replace by cartilage (also in  Fig2).

L245 : Replace « metformin ameliorates OA progression by supressing «  by « Metformin slows OA progression by reducing » 

Figure 2 : L249 : Review the sentence. Precise concentration of MIA and metformin as in fig 1.

L255 : number of rats ?  3 independant experiments of 3 rats ? or more rats as in  fig1

L265 : «  IL1-stimulated human OA chondrocytes » rather than « IL-1-stimulated OA chondrocytes of human »

Figure 3 : L276 : Review the sentence.

 L278 : « Counting of positive cells…is shown », rather than « positive cells …are shown ».

L279-280 ; Move (B) before « The human chondrocytes…

L281 : Move « the data are reported…..  before (B) ; Give the number of experiments for mRNA levels.

The legends of the abscissa in B, C and D are illegible

L289 : Give the role of MKLL.

Why did you increase the concentration of metformin in human OA chondrocytes ?

Explain the choice of IHC with caspase 1 rather than with caspase 1

Did you do a western blot with p-MLK ?

Did you do a western blot of total AMPK because metformin also increase total AMPK in the litterature. Talk about it in the discussion

Figure 4 : L297 : it’s : « Counting of positive cells… »

L298-300 : Review the sentence.

Give the number of experiments, for each panel A and B : number of rats ; number of WB…

L305 : Quantify the fluorescence to attest the results of fig 5A and B.

Figure 5 :

Explain «  Nil « at the left of the panels ;

L312 : Review the sentence : confocal microscopy for markers of autophagolysosome.

It’s not the expression of BODIPY, but of cytoplasmic proteins ?  Add the nucleus staining in the legend.

Add a space after LAMP1

Give the number of experiments

L305-307 : The paragraph « In addition, patients…fig6 »  should not appear here, because it is not related to autophagy. And you realized IHC with rat samples not with patients !

Figure 6 : L318 : Review the sentence. Give the number of rats par group

L327 : I don’t understand the groups : You have diabetic and non diabetic patients, so you could have a role of diabete itself on OA, rather than a role of metformin on progression of OA. So the conclusion was not valid.

Which NSAIDs are you talking about ?

What is the age and gender distribution of patients? Can you make sub groups?

Fig 7 : L336 : Give the number of patients in each group.

Redefine groups : « non diabetic vs diabetic with metformin » , rather than « non-met vs met »

Discussion

L340 : Little is known regarding its potential therapeutic effect in OA » . I do not agree. At least 10 articles on the effect of met on OA !!

Three references must be exploited : ref 29+ Li 2020 https://doi.org/10.1186/s13075-020-2129-y. + Lu https://doi.org/10.1371/ journal.pone.0191242.

L 343 : The present result confirm but not demonstrate its potential clinical use in OA patients

L344 : Replace « met ameliorates experimental OA by suppressing « by « met slows OA progression by reducing »

L354 : Add diabetic in « in the knees of diabetic OA patients »

L356 : « The decrease of pro-inflammatory cytokines », but not « the suppression of pro-inflammatory cytokines »

L365 : Met is not as an agonist of AMPK !;; It induces AMPK signalling . ref13 was not adapted, but cite ref 29 and Wang 2020 and Zhang 2020.

-Wang, Chenzhong, et al. « Metformin Mitigates Cartilage Degradation by Activating AMPK/SIRT1-Mediated Autophagy in a Mouse Osteoarthritis Model ». Frontiers in Pharmacology 11 (2020): 1114. https://doi.org/10.3389/fphar.2020.01114.

-Zhang, Mengqi, Yaping Liu, Zhikun Huan, Yan Wang, et Jin Xu. « Metformin Protects Chondrocytes against IL-1β Induced Injury by Regulation of the AMPK/NF-κ B Signaling Pathway ». Die Pharmazie 75, no 12 (1 décembre 2020): 632‑36. https://doi.org/10.1691/ph.2020.0762.

L370 : « We found that… « Stop the sentence after TIMPs, because here, you did not work on Th17/Treg population. Modify the sentence and add ref 11-12.

Discuss the anti-inflammatory role of metformin in relation with your previous results and others : Park et al 2019, Zhang et al, 2020

L381 : Replace «  Met treatment ameliorated pain »  by « met decreased pain »

L386 : You must discuss your results of autophagy in relation with results of others :

-Wang, Chenzhong, et al. « Protective Effects of -Metformin against Osteoarthritis through Upregulation of SIRT3-Mediated PINK1/Parkin-Dependent Mitophagy in Primary Chondrocytes ». Bioscience Trends 12, no 6 (22 janvier 2019): 605‑12. https://doi.org/10.5582/bst.2018.01263.

-Wang, Chenzhong, et al. « Metformin Mitigates Cartilage Degradation by Activating AMPK/SIRT1-Mediated Autophagy in a Mouse Osteoarthritis Model ». Frontiers in Pharmacology 11 (2020): 1114.

L393 : « OA with DM patient groug ». You mean DMM ? (destabilisation of the meniscus ?) Specify the abbreviation and replace patient group by animal group ! mice or rats ! review the sentence. It is not clear.

L369 : Review paragraph  because you mixed DM patients rather than diabetic patients or OA patients, DMM rats ?, non DM-patients rather than non met-patients ? It’s incomprehensible.

L402 : e..using both Met and clecoxib e: Refer to previous results: Lu, Chieh-Hua, , et al. « Combination COX-2 Inhibitor and Metformin Attenuate Rate of Joint Replacement in Osteoarthritis with Diabetes: A Nationwide, Retrospective, Matched-Cohort Study in Taiwan ». PloS One 13, no 1 (2018): e0191242. https://doi.org/10.1371/journal.pone.0191242.

Conclusions :

L415. Split the sentence into 2 sentences

L418 : Specify « group in MIA-inducced OA rats »

References :

Ref 15 : Replace poster by article Latourte A. :

Latourte, Augustin, Chahrazad Cherifi, Jérémy Maillet, Hang-Korng Ea, Wafa Bouaziz, Thomas Funck-Brentano, Martine Cohen-Solal, Eric Hay, et Pascal Richette. « Systemic Inhibition of IL-6/Stat3 Signalling Protects against Experimental Osteoarthritis ». Annals of the Rheumatic Diseases 76, no 4 (1 avril 2017): 748‑55. https://doi.org/10.1136/annrheumdis-2016-209757.

Author Response

Thank you for your comments and pointing out. We have addressed your comment and concerns by revising the manuscript. The point-by-point replies are given in this manuscript. The sentence is corrected of manuscript, and the corrected sentences mark to red. In addition, we include recently finding from studies of metformin and osteoarthritis in manuscript.

We hope that we have addressed satisfactorily all concerns raised by the reviewers, and that this manuscript is now suitable for publication. Thank you again for your comments.

Reviewer 2 Report

Dr. Na et al. reported that Metformin improves monosodium-iodoacetate-induced osteoarthritis via regulation of pain mediators and autophagy-lysosomal pathway. Although there are several reports on this topic, this study is well designed and written as well as has enough new information that is merited it to be published in Cells.

Author Response

Thank you for your comments. We hope that we have addressed satisfactorily all concerns raised by the reviewers, and that this manuscript is now suitable for publication. Thank you again for your comments.

Reviewer 3 Report

Authors show very interestingly that Metformin improves monosodium-iodoacetate-induced osteoarthritis (OA) via regulation of pain mediators and autophagy-lysosomal pathway. Interestingly, their results imply the possibility of the therapeutic use of metformin in OA patients, based on its ability to suppress pain and protect cartilage.

They use rats and chondrocytes from OA patients, which are appropriate models for the study.

However they said that the autophagy-lysosomal pathway regulates the process, but the characterization of the autophagy pathway that they made is not enough to say this. They only show immunofluorescence images of lc3 and lamp separately. To say that autophagy is increased is necessary to show increased autophagolysosomes (increase fusion of lc3 and lamp). Moreover, the authors have also to make complementary experiments to show increased autophagy markers expression, as western blot to quantify lc3-I to lc3-II conversion, and p62 expression. Moreover, an autophagy inhibitor (usually Bafylomicin or Clororquine) is also required in-vitro to confirm that your drug (metformin) increased autophagy in your context.

Moreover, authors have to better explain some points in the material and methods; otherwise, experiments will not be reproducible. Globally, data presented are interesting and promising from a therapeutic point of view, but the paper cannot be accepted if the above-mentioned improvements will not be made.

Abstract= is badly written. For example in the methods are not indicated all the methods used. Celecoxib is mentioned at the end of the abstract without any link to other parts.

lane 39= celecoxib was not mentioned above. Is not clear why it is mentioned at this point

introduction= you have to introduce the use of celecoxib

lane 101= indicate the number of rats used for every experimental groups and their sex. Discuss the potential involvement of sex in the results if groups are not homogeneous.

lane 107= write how the oral administration is made (in the food, water….?)

lane 142= indicates the instrument used for make sections

lane 145= write the code of every antibody

lane 165= write the protocol used for the real-time PCR amplification

lane 181= write the protocol of protein extraction from samples. Indicate the antibodies code. Write the protocol for the quantification of the obtained western blot bands.

lane 220= the protocol of immunofluorescence is missing and also the code of the antibodies.

figure1= letter b and d are missing in the figure. The last image of cross-section of the spinal cord is not explained in the caption.

lane 241 = “Safranin O staining of the femur revealed that metformin decreased cartilage destruction of the knee joint (Figure 2C). Additionally, metformin administration reduced both the OARSI and the total Mankin score compared to the control group (Figure 2D).”

You have to explain safranin staining and OARSI and Mankin score in the methods.

Fig. 2C= better explanation of the figure is needed. What the images up and down represent?

Lane 289= levels of pAMPK are normalized to the level of total AMPK? If yes you have to specify this in the text, otherwise it is necessary or you cannot say that phosphorylation is increased, because different levels of not phosphorylated AMPK may be. You have to show also western blot for AMPK not phosphorylated.

Lane 301= why do you use bodipy? It doesn't seem necessary for your study. If it is, you must explain why you use it. Even in materials and methods. A quantification of lc3 and lamp is necessary, you can quantify the images of different experiments and associate the images with western blots. Furthermore, to verify that autophagy is increased, it is necessary to evaluate the colocalization of LC3 and lamp, to verify the fusion between autophagosome and lysosome.

Fig. 5= how many experiments? A quantification is needed

Figure 6= caption is not written properly.

Is not clear if you use 6 animal per group how is possible that the graph results from only 3 independent experiments. And why you use SD instead of SEM? You have to better explain these points.

3.7= explain how is KL grade quantified

Discussion= you have to comment about the effect of autophagy activation in relation to the therapeutic effects of metformin in OA and in relation to downregulation of pain mediators.

Lane 415= you have to show increased colocalization of lc3 and lamp to say that there is: “induced autophagosome-lysosome fusion phenotype.”

lane 417= haw celecoxib works? Discuss this and the possible positive effect of the combination therapy.

Author Response

(The authors gave the same response as above.)

Round 2

Reviewer 1 Report

The major concerns are on the English language of the revised form and the lack of rigor in the corrections.

Introduction :

L75-89 : the authors have completed the bibliography on the effects of metformin and OA but english language  is not good :

L75« Recently suggest that the metformin as noble therapy for osteoarthritis. « 

Replace by : « Recent studies suggest the potential use of metformin for OA therapy. »

L76 « The metformin attenuated osteoarthritis with destabilization of the medial meniscus (DMM)-operation mice through cartilage damage and pain control[14]. »

Replace by : « Metformin attenuated cartilage damage and controlled pain in a destabilization of the medial meniscus (DMM) model of OA in mice  [14]» 

L78  «  Moreover metformin reduce inflammatory response of chondrocytes by IL-1β, which is protecting injury cell death[15]. »

Replace by  « Moreover, metformin protected chondrocytes against IL-1β-induced inflammatory response and injury [15] »

L80 «  In our study also, we shows that therapeutic effect of meformin. « 

Replace by « We also showed therapeutic effects of metformin in a previous study [16]. »

L81 « The metformin treated Ad-hMSCs increased antinociceptive activity and chondroprotective effect in OA rat[16]. « 

« OA rats treated with metformin-stimulated adipose tissue-derived human MSCs (Ad-hMSCs) increased antinociceptive activity and chondroprotective effects. »

L85-87. « In cohort study, the celecoxib and metformin combination treated OA patients with type 2 diabetes mellitus were lowered joint replacement surgery rates. » Review the sentence.  « … have lowered joint replacement surgery rates ? »

METHODS :

L122 : Why did you change the dose of metformin (50 mg/kg)  in the metormin/celecoxib combination?

Response: thank you for your comment. Metformin alone was 100mg/kg, but in combination with celecoxib, the lower dose of Metformin, but 50mg/kg, which had an effect, was used. Therefore, 50mg/kg and celecoxib were administered in combination with a dose lower than the Metformin dose used alone. Thus, the effect was observed.

OK but  reschedule the celecoxib dose

L160 Replace sentence by « Stained samples were analysed by… »

L182 : Give the approval number for human cartilage samples. You did not respond to the request :

You need the ethical approval  number. What is this number UC18RESi0038 ? (the same in 2.12. Grading system in OA patients) !

L184 : « Finally, chondrocytes were incubated in Dulbecco’s… », not « finally, in chondrocytes were incubated Dulbecco’s »

L185-187 : Remove the duplicate sentence in black. Did you use 2 doses of IL-1 : 10 or/and 20 ng/ml ?

L238.Immunofluorescence again : As you explain, Bodipy is not a specific marker of autophagosome, then modify the sentence as follow : « The cells were stained for lipid droplets and with specific antibodies for autophagosome ..»    

L247 : Grading system with a capital letter

Results :

Figure 1 : L273-274 : Review the sentence again (after the title) « The rats were injected with 3 mg of MIA. After OA induction, the rats were administrated with met », not «  the rats were adminstration with met »

Specify the duration of metformin administration, one orally administration or during 15 days ?

Specify in the method if Met is in drinking water or force-feeding

Figure 2 : L294 : Review the sentence again as above.

Moreover L295:  Modify the sentence as follow : The rats were sacrified at day 15 after OA induction to collect joint tissues.

Figure 3 : L323-326 : Review the sentence again as above

The legends of the abcissas in B, C and D are always illegible !

L345 : You add the sentences :   «  In addition, the inflamation cell death factor, caspase-1 was increased by hydroxycholoroquin(HCQ) which is bloker to the autophage and  lysosome formation. These result suggest that the metformin regulate inflammation cell death through autophagy formation in OA chondrocytes. » Review the english language.

Figure 3 : L324-325 : Review the sentence as above.

 The legends of the abscissa in B, C and D are always illegible

Figure 4 : Enlarge the figure

Figure 5 : Explain «  Nil » in the legend; replace BADIPY by BODIPY in the legend. Space after « (red).  »

Figure 6 : L318 : Review the sentence again. : « the rats were oral administrated metformin and celecoxib » ?

L369-370 : Give numbers to references

L372-375 : Review english language (« both of, on both of… »). Compare metformin group to the combination of met and celecoxib.

L396 : Remove «  NSAIDs to »

L394 ; Chapter 3-8, not 3-7

Fig 7 : L336 : Give the number of patients in each group as previously asked, in the legend

Indicate the gendre of patients in the mat-meth

Discussion

L414 : Delete « rat model »

L415 : You did not replace « met ameliorates experimental OA by suppressing « by « met slows OA progression by reducing »

L425. The sentence is not correctly corrected : « a lower KL grade in the knees of diabetic OA patients » not « a lower KL grade diabetic in the knees of OA patients »

L427 : You did not replace « The decrease of pro-inflammatory cytokines », but not « the suppression of pro-inflammatory cytokines »

L435 : Met induces AMPK signalling, not AMPK expression.  Add signalling : « metformin, as an inducer of AMPK signalling, »

L441 : Modify the sentence (english language) : « moreover,…[15] ».

Modify the other sentence as follows: « Our previous study also shows that metformin has a therapeutic effect in experimental OA [16]. »

L466 « The inflammatory… response ». Review english language

L469 : « recently, …chondrocytes [43] » Review english language

L463 : Autophagy :  ref 40 is not adapted (curcumin and autophagy): replace by (as previously suggested)

-Wang, Chenzhong, et al. « Protective Effects of -Metformin against Osteoarthritis through Upregulation of SIRT3-Mediated PINK1/Parkin-Dependent Mitophagy in Primary Chondrocytes ». Bioscience Trends 12, no 6 (22 janvier 2019): 605‑12. https://doi.org/10.5582/bst.2018.01263.

(Leave the ref 41)

L468 : review paragraph : Remove the capital letter from "The effect" ; delete « there was »

Add ref 18 to the end of the paragraph.

Conclusions :

L510 : Remove « in MIA-inducced OA rats » at the end of the sentence.

Author Response

(The authors gave the same response as above.)

Reviewer 3 Report

Overall, authors answer my previous comments and the paper is appropriate for "cells", but English and statistics must be adjusted. 

Author Response

(The authors gave the same response as above.)
